

# Associations of obesity-related indices with mild cognitive impairment in adults 60 years and older with type 2 diabetes: a retrospective study

Jing Feng[1,2,3], Zhenjie Teng[4] and Shuchun Chen[1,2,3]

[1] Department of Endocrinology, Hebei Medical University, Shijiazhuang, China
[2] Department of Endocrinology, Hebei General Hospital, Shijiazhuang, China
[3] Hebei Key Laboratory of Metabolic Disease, Shijiazhuang, China
[4] Department of Neurology, Hebei General Hospital, Shijiazhuang, China

Corresponding author
Shuchun Chen,
chenshuc2014@163.com

## ABSTRACT

**Objective.** To investigate the relation between obesity-related indices and mild cognitive impairment (MCI) in elderly patients with type 2 diabetes (T2D).

**Methods.** A total of 597 eligible elderly patients with T2D were included in this retrospective study. All patients were divided into MCI group and normal cognitive group based on neuropsychological assessment. Twelve obesity-related indices were calculated, including body mass index (BMI), waist-hip ratio (WHR), waist-to-height ratio (WHtR), lipid accumulation product (LAP), body roundness index (BRI), conicity index (CI), visceral adiposity index (VAI), body adiposity index (BAI), abdominal volume index (AVI), a body shape index (ABSI), triglyceride glucose (TyG) index and cardiometabolic index (CMI). Multivariate logistic regression analysis, tests for trend and restricted cubic splines were used to assess the relationships between the tests for trend and MCI in elderly patients with T2D. Receiver operating characteristic (ROC) curves and areas under the curves (AUC) were used to assess the performance and predictive ability of the obesity-related indices for identifying MCI in elderly patients with T2D.

**Results.** Multivariate logistic regression showed that elevated BMI, WHR, WHtR, LAP, BRI, CI, VAI, AVI, TyG index, and CMI were associated with an increased risk of MCI in elderly T2D patients after adjusting for potential confounders (all $P < 0.05$). In addition, TyG index, LAP, CMI, VAI, AVI, WHR, WHtR, BRI, and CI had negative correlations with Montreal Cognitive Assessment (MoCA) scores (all $P < 0.05$). There was a significant linear trend between the levels of BMI ($P$ for trend $= 0.004$, $P$ for non-linearity $= 0.637$), WHR ($P$ for trend $= 0.006$, $P$ for non-linearity $= 0.430$), WHtR ($P$ for trend $< 0.001$, $P$ for non-linearity $= 0.452$), BRI ($P$ for trend $< 0.001$, $P$ for non-linearity $= 0.252$), AVI ($P$ for trend $< 0.001$, $P$ for non-linearity $= 0.944$), and TyG index ($P$ for trend $< 0.001$, $P$ for non-linearity $= 0.514$) and risk of MCI in elderly patients with T2D after adjusting for potential confounders. There was a nonlinear association between LAP, VAI or CMI and risk of MCI in elderly patients with T2D (all $P$ for non-linearity $< 0.001$). CMI had the greatest AUC (AUC $= 0.682$), followed by VAI (AUC $= 0.679$), TyG index (AUC $= 0.673$), LAP (AUC $= 0.669$), AVI (AUC $= 0.580$), WHtR and BRI (AUC $= 0.575$), BMI (AUC $= 0.560$), CI (AUC $= 0.556$), WHR (AUC $= 0.554$), BAI (AUC $= 0.547$), and ABSI (AUC $= 0.536$).

**Conclusion**. Elevated obesity-related indices, particularly CMI, VAI, TyG index and LAP, which displayed the higher predictive power, were instrumental in forecasting and evaluating MCI in elderly T2D patients. These findings may provide clues for future studies exploring early diagnostic biomarkers and treatment of MCI in elderly T2D patients.

## INTRODUCTION

Type 2 diabetes (T2D) and cognitive impairment are among the most prevalent chronic conditions in elderly populations, posing significant challenges to global public health (*Srikanth et al., 2020*; *Luo et al., 2022*). Cognitive impairment is increasingly acknowledged as a common yet often underrecognized complication of T2D (*Srikanth et al., 2020*; *van Sloten et al., 2020*). Epidemiological studies have demonstrated that individuals with T2D face a heightened risk of cognitive decline, particularly in older adults (*Biessels et al., 2006*; *Xue et al., 2019*). It is estimated that approximately 20% of individuals aged 60 and older with T2D may develop dementia, which represents an advanced stage of cognitive impairment (*Srikanth et al., 2020*; *Bunn et al., 2014*). Specifically, T2D is associated with an increased likelihood of developing mild cognitive impairment (MCI) (up to 45%) (*You et al., 2021*) and accelerated cognitive decline from MCI to dementia (*Srikanth et al., 2020*; *Biessels & Whitmer, 2020*; *Xu et al., 2010*). Furthermore, cognitive impairment, especially dementia, adversely affects multiple aspects of daily life for individuals with T2D, further exacerbating cognitive decline (*Srikanth et al., 2020*; *Luo et al., 2022*). Consequently, there is an urgent need to enhance our understanding of effective diagnostic markers for MCI, an early stage of cognitive impairment, in patients with T2D.

Obesity represents a prevalent global health challenge (*NCD Risk Factor Collaboration, 2024*) and is linked to numerous age-related conditions and diseases, including T2D, cerebrovascular disease, and cognitive impairment (*Lin & Li, 2021*; *Buie et al., 2019*; *Barrea et al., 2023*; *Livingston et al., 2024*). A comprehensive analysis of 21 cohort studies involving 5,060,687 participants demonstrated that obesity, as quantified by waist circumference (WC), is significantly linked to an elevated risk of cognitive impairment (*Tang et al., 2021*). Moreover, additional research has identified obesity as a significant risk factor for MCI among elderly populations (*Yao et al., 2016*). Although the exact mechanisms linking obesity to cognitive impairment are not yet fully understood, several potential pathways have been suggested. These include cerebral microvascular dysfunction, oxidative stress, endothelial dysfunction, blood–brain barrier disruption, neuroinflammation, insulin resistance, and gut microbiota dysbiosis (*Buie et al., 2019*; *Henn et al., 2022*; *Zheng et al., 2024*; *Balasubramanian et al., 2021*).

Obesity-related indices, which are convenient and accessible measures, serve as surrogate markers for various types of obesity. Previous research has documented associations

between elevated obesity-related indices, such as high waist-hip ratio (WHR) (*Li et al., 2018*) or triglyceride glucose (TyG) index (*Teng et al., 2022*), and the risk of cognitive impairment in patients with T2D. Furthermore, *Abi et al. (2022)* proposed that diabetes may significantly contribute to the adverse effects of obesity on cognitive function. However, limited research has examined the associations between obesity-related indices and MCI in elderly T2D patients. This study aims to evaluate whether obesity-related indices are associated with an increased risk of MCI in elderly patients with T2D.

## MATERIALS & METHODS

### Study population

This retrospective study encompassed patients aged 60 years and older with T2D, who were admitted to Hebei General Hospital between January 2021 and November 2023 according to medical records. The diagnosis of T2D was established according to the criteria set forth by the American Diabetes Association (*ElSayed et al., 2023*). Exclusion criteria included: (1) patients diagnosed with type 1 diabetes mellitus; (2) patients lacking cognitive function assessments; (3) patients meeting the diagnostic criteria for dementia (*Ismail et al., 2020*); (4) patients experiencing cerebrovascular events, such as stroke, within three months prior to the study; (5) patients with other specific conditions that could influence cognitive function assessment, including anxiety, depression, brain injuries, or carbon monoxide poisoning. Ultimately, 587 elderly patients were included in our study. The authors did not have access to any information that could have identified individual participants either during or after the data collection process. The procedure for patient selection is illustrated in Fig. 1.

### Ethical approval

This study followed the principles in the Declaration of Helsinki and was approved by the Ethical Committees of Hebei General Hospital (NO.2024-LW-109). The informed consent was waived because the study was a retrospective analysis and the data of all participants were anonymized.

### Data collection

The characteristics of all patients were meticulously documented, encompassing age, body height (BH), body weight (BW), WC, hip circumference (HC), years of education, smoking and alcohol status, duration of T2D, anti-diabetic drugs, and history of stroke, hypertension, and coronary heart disease (CHD). Laboratory parameters including fasting plasma glucose (FPG), glycosylated hemoglobin (HbA1c), total cholesterol (TC), triglycerides (TG), low-density lipoprotein cholesterol (LDL-C), high-density lipoprotein cholesterol (HDL-C), blood urea nitrogen (BUN), serum creatinine (Scr), uric acid (UA), and estimated glomerular filtration rate (eGFR) were quantitatively analyzed through automated biochemical analyzer and glycosylated hemoglobin analyzer platforms in accordance with established clinical laboratory procedures.

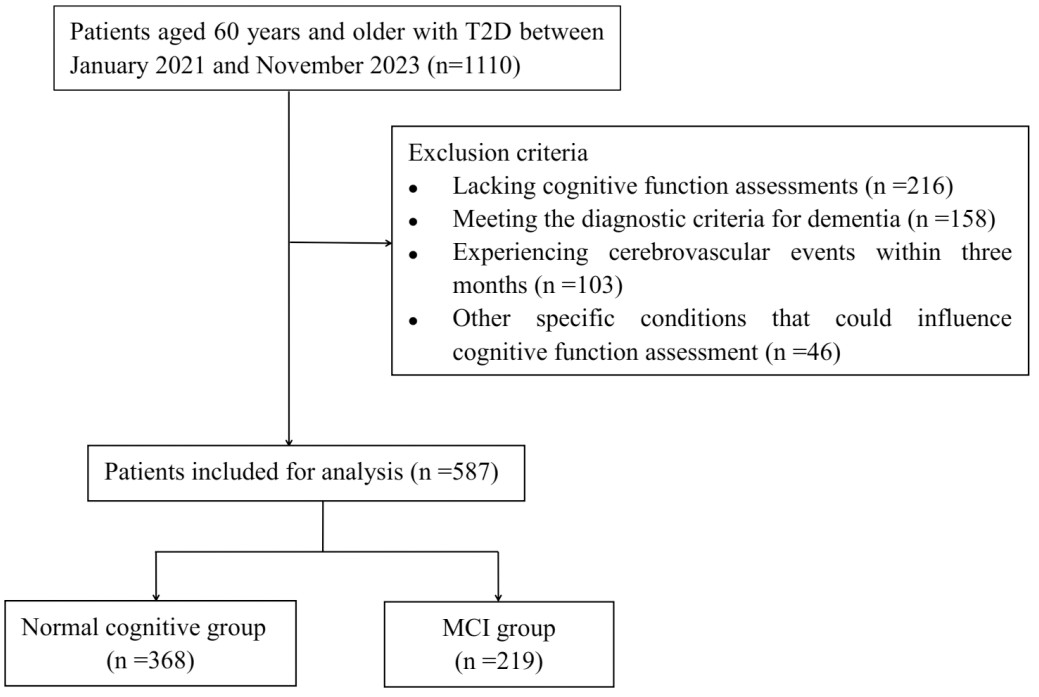

**Figure 1** Flowchart of patient selection.

## Calculation of obesity-related indices

The obesity-related indices were calculated (*Huang et al., 2022*; *Zhang et al., 2024*; *Ou et al., 2021*) as follows.

1. Body mass index (BMI) = BW(kg)/BH$^2$(m)
2. WHR = WC(cm)/HC(cm)
3. Waist-to-height ratio (WHtR) = WC(cm)/BH(cm)
4. Lipid accumulation product (LAP) = (WC(cm)–65)×TG(mmol/L) in males and
   LAP = (WC(cm)–58)×TG(mmol/L) in females
5. Body roundness index (BRI) = $364.2 - 365.5 \times \sqrt{1 - (\frac{WC(m)}{\pi \times BH(m)})^2}$
6. Conicity index (CI) = $\frac{WC(m)}{0.109 \times \sqrt{\frac{BW(kg)}{BH(m)}}}$
7. Visceral adiposity index (VAI) = $\left(\frac{WC(cm)}{39.68 + (1.88 \times BMI)}\right) \times \left(\frac{TG(mmol/L)}{1.03}\right) \times \left(\frac{1.31}{HDL - C(mmol/L)}\right)$
   in males and
   VAI = $\left(\frac{WC(cm)}{36.58 + (1.89 \times BMI)}\right) \times \left(\frac{TG(mmol/L)}{0.81}\right) \times \left(\frac{152}{HDL - C(mmol/L)}\right)$ in females
8. Body adiposity index (BAI) = $\frac{HC(cm)}{BH(m)^{3/2}} - 18$
9. Abdominal volume index (AVI) = $\frac{2 \times WC(cm)^2 + 0.7 \times (WC(cm) - HC(cm))^2}{1000}$
10. Abody shape index (ABSI) = $\frac{WC(m)}{BMI(kg/m^2)^{2/3} \times BH(m)^{1/2}}$
11. TyG = ln [TG (mg/dl)×FPG (mg/dl)/2]
12. Cardiometabolic index (CMI) = TG/HDL-C×WHtR

## Evaluation of cognitive function

The Montreal Cognitive Assessment (MoCA) was utilized to evaluate cognitive function in all participants. Cognitive impairment was evaluated using objective criteria based on MoCA scores: a score of ≤13 indicated impairment in individuals with no formal education, ≤19 for those with 1–6 years of education, and ≤24 for individuals with 7 or more years of education (*Lu et al., 2011*). According to the established criteria (*Albert et al., 2011*), patients enrolled in our study were categorized into either the normal cognitive group and MCI group.

## Statistical analysis

Continuous variables adhering to a normal distribution were presented as mean (standard deviation) and analyzed using an independent samples $t$-test. For continuous variables that did not follow a normal distribution, the median (interquartile range) was utilized for description, with statistical significance evaluated *via* the Mann–Whitney U test. Categorical variables were summarized as frequency (percentage) and examined using chi-square tests. Logistic regression models were utilized to examine the association between obesity-related indices and MCI in elderly patients with T2D. Tests for trend were conducted to evaluate the dose-dependence relationships between obesity-related indices and MCI. Spearman correlation analysis was performed to assess the correlations between the MoCA scores and obesity-related indices. To further investigate potential non-linear relationships, restricted cubic splines were employed. The performance and predictive capability of obesity-related indices in identifying MCI among elderly patients with T2D were assessed using receiver operating characteristic (ROC) curves and the area under the curve (AUC). Statistical analyses were performed using SPSS version 26 (IBM Corporation, Armonk, NY) and the R programming environment (version 4.2.3, the R Foundation for Statistical Computing, Vienna, Austria). $P < 0.05$ was considered statistically significant.

# RESULTS

## Participants characteristics

A total of 587 patients with T2D (368 in the normal cognitive group and 219 in the MCI group) were included in our study. Table 1 shows the characteristics of patients in both groups. Patients with MCI were significantly older and had lower education levels than those in the normal cognitive group ($P < 0.05$). Patients in MCI group had lower frequencies of glucagon-like peptide-1 receptor (GLP-1R) agonists and sodium-dependent glucose transporters 2 (SGLT-2) inhibitors usage than those in the normal cognitive group ($P < 0.05$). In addition, compared with normal cognitive group, the MCI group presented higher FPG, HbA1c and TG, but lower HDL-C ($P < 0.05$).

## Correlations between obesity-related indices and MoCA scores

As illustrated in Fig. 2, the TyG index ($r = -0.22$, $P < 0.001$), LAP ($r = -0.20$, $P < 0.001$), CMI ($r = -0.19$, $P < 0.001$), and VAI ($r = -0.19$, $P < 0.001$) exhibited significantly stronger negative correlations with MoCA scores compared to other obesity-related indices such as AVI, WHR, WHtR, BRI, and CI. Notably, There were no significant correlations between BAI, BMI, ABSI, and MoCA scores ($P > 0.05$).

**Table 1 Clinical and laboratory characteristics of participants with T2D between normal cognitive group and MCI group.**

| Variable | Total ($n = 587$) | Normal cognitive group ($n = 368$) | MCI group ($n = 219$) | P value |
|---|---|---|---|---|
| Age (years) | 67 (64–72) | 67 (63–71) | 69 (64–74) | 0.007[*] |
| Gender (Male, %) | 300 (51.1) | 196 (53.3) | 104 (47.5) | 0.176 |
| Education (years) | 9 (9–12) | 12 (9–12) | 9 (6–12) | 0.001[*] |
| Hypertension, n(%) | 365 (62.2) | 219 (59.5) | 146 (66.7) | 0.084 |
| CHD, n(%) | 131 (22.3) | 79 (21.5) | 52 (23.7) | 0.522 |
| Stroke, n(%) | 116 (19.8) | 64 (17.4) | 52 (23.7) | 0.062 |
| Smoking, n(%) | 91 (15.5) | 56 (15.2) | 35 (16.0) | 0.805 |
| Alcohol use, n(%) | 65 (11.1) | 43 (11.7) | 22 (10.0) | 0.541 |
| Duration of T2D (years) | 10 (4–20) | 11 (5–20) | 10 (4–19) | 0.119 |
| Anti-diabetic drugs, n(%) | | | | |
| Metformin | 391 (66.6) | 241 (65.5) | 150 (68.5) | 0.455 |
| Insulin | 305 (52.0) | 198 (53.8) | 107 (48.9) | 0.246 |
| Sulfonylureas | 170 (29.0) | 108 (29.3) | 62 (28.3) | 0.789 |
| Nateglinide | 45 (7.7) | 31 (8.4) | 14 (6.4) | 0.371 |
| Thiazolidinediones | 28 (4.8) | 18 (4.8) | 10 (4.6) | 0.858 |
| Glycosidase inhibitors | 247 (42.1) | 154 (41.8) | 93 (42.5) | 0.883 |
| DPP-4 inhibitors | 71 (12.1) | 47 (12.8) | 24 (11.0) | 0.515 |
| GLP-1R agonists | 89 (15.2) | 68 (18.5) | 21 (9.6) | 0.004[*] |
| SGLT-2 inhibitors | 69 (11.8) | 55 (14.9) | 14 (6.4) | 0.002[*] |
| FPG (mmol/L) | 7.82 (6.20–10.49) | 7.49 (5.97–10.17) | 8.77 (6.81–11.16) | 0.001[*] |
| HbA1c (%) | 8.6 (7.2–10.2) | 8.2 (7.1–10.1) | 8.9 (7.7–10.8) | 0.001[*] |
| TC (mmol/L) | 4.47 (3.61–5.29) | 4.44 (3.57–5.27) | 4.51 (3.68–5.44) | 0.178 |
| TG (mmol/L) | 1.27 (0.93–1.93) | 1.12 (0.85–1.68) | 1.59 (1.14–2.35) | <0.001[*] |
| HDL-C (mmol/L) | 1.14 (0.95–1.35) | 1.18 (0.98–1.39) | 1.06 (0.91–1.29) | <0.001[*] |
| LDL-C (mmol/L) | 2.89 (2.20–3.52) | 2.82 (2.16–3.50) | 2.95 (2.25–3.55) | 0.182 |
| BUN (mmol/L) | 5.80 (4.80–7.00) | 5.80 (4.71–7.00) | 5.80 (4.80–7.00) | 0.882 |
| Scr (μmol/L) | 64.5 (55.9–76.5) | 65.2 (56.1–76.2) | 63.7 (55.4–76.6) | 0.376 |
| UA (μmol/L) | 296 (242–367) | 294 (240–362) | 303 (247–372) | 0.278 |
| eGFR (ml/min/1.73 m$^2$) | 91.1 (81.8–97.6) | 91.6 (81.9–97.8) | 90.5 (81.5–96.8) | 0.350 |

**Notes.**

[*]Denotes significance at a P value of < 0.05.

Abbreviations: T2D, type 2 diabetes; MCI, mild cognitive impairment; CHD, coronary heart disease; DPP-4, dipeptidyl peptidase-4; GLP-1R, glucagon-like peptide-1 receptor; SGLT-2, sodium-dependent glucose transporters 2; FPG, fasting plasma glucose; HbA1c, glycosylated hemoglobin; TC, total cholesterol; TG, triglyceride; HDL-C, high density lipoprotein cholesterol; LDL-C, low density lipoprotein cholesterol; BUN, blood urea nitrogen; Scr, serum creatinine; UA, uric acid; eGFR, estimated glomerular filtration rate.

## Associations between obesity-related indices and MCI

Compared to T2D patients with normal cognition, those with MCI exhibited significantly higher values in BMI, WHR, WHtR, LAP, BRI, VAI, CI, AVI, TyG index, and CMI ($P < 0.05$) (Table 2). The results of logistic regression analyses are shown in Table 3. In the unadjusted model (**Model 1**), obesity-related indices, excluding WHR, BAI, and ABSI, were associated with MCI in elderly T2D patients ($P < 0.05$). After adjusting for age,

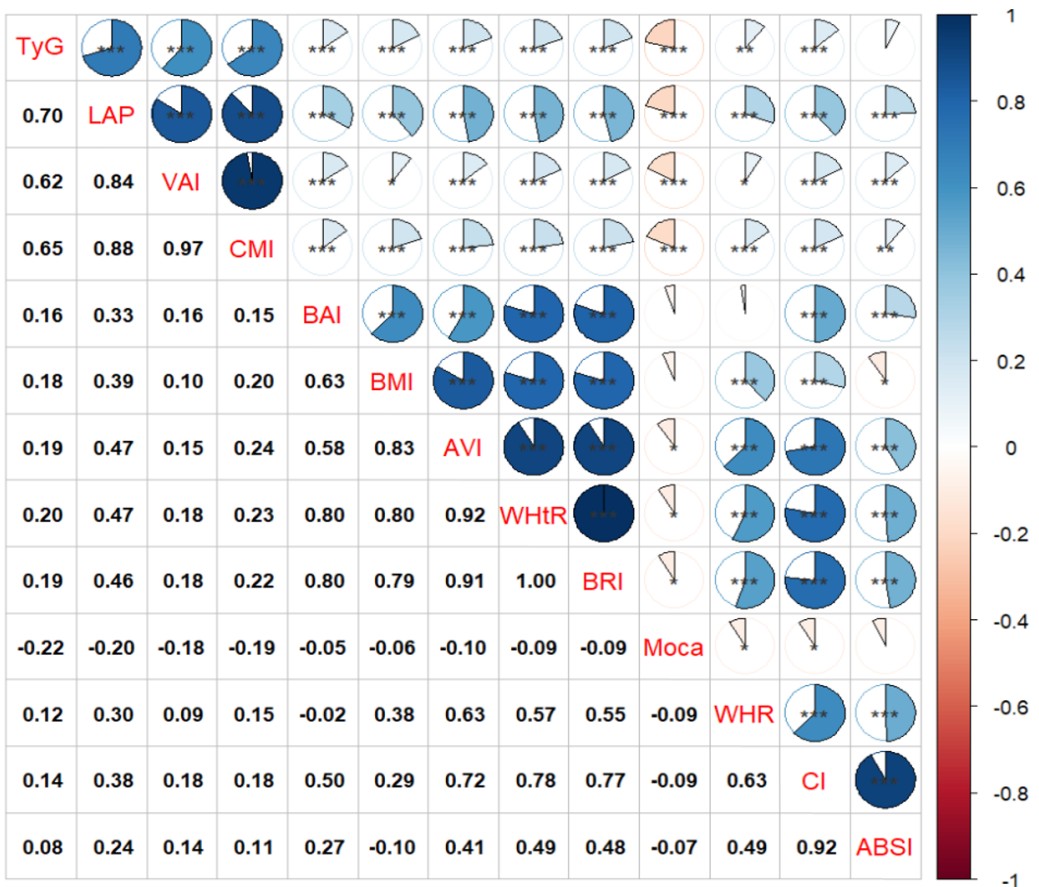

**Figure 2** **Associations between obesity-related indices and MoCA score.** * $P < 0.05$, ** $P < 0.01$, *** $P < 0.001$.

education, hypertension, stroke, anti-diabetic drugs, and HbA1c (**Model 3**), elevated BMI, WHR, WHtR, LAP, BRI, CI, VAI, AVI, TyG index, and CMI were significantly associated with an increased risk of MCI in elderly T2D patients (all $P < 0.05$).

The levels of obesity-related indices, excluding CI, BAI, and ABSI, were positively associated with the risk of MCI in elderly T2D patients (Table 4). Relative to the lowest quartile, the odds ratios (ORs) for the highest quartile were as follows: BMI had an OR of 1.983 (95% confidence interval (CI) [1.187–3.312], $P$ for trend = 0.004); WHR had an OR of 1.519 (95% CI [0.918–2.513], $P$ for trend = 0.037); WHtR had an OR of 1.903 (95% CI [1.140–3.177], $P$ for trend = 0.006); LAP had an OR of 5.743 (95% CI [3.282–10.050], $P$ for trend < 0.001); BRI had an OR of 1.939 (95% CI [1.159–3.245], $P$ for trend = 0.006); VAI had an OR of 5.456 (95% CI [3.125–9.443], $P$ for trend < 0.001); AVI had an OR of 2.734 (95% CI [1.623–4.605], $P$ for trend < 0.001); TyG index had an OR of 7.440 (95% CI [4.139–13.373], $P$ for trend < 0.001); and CMI had an OR of 7.355 (95% CI [4.134–13.085], $P$ for trend < 0.001).

After adjusting for age, education, hypertension, stroke, anti-diabetic drugs (GLP-1R agonists and SGLT-2 inhibitors), and HbA1c, there was a linear trend between BMI ($P$ for

**Table 2** Comparison of the obesity-related indices between normal cognitive group and MCI group in participants with T2D.

| Variable | Total (*n* = 587) | Normal cognitive group (*n* = 368) | MCI group (*n* = 219) | *P* value |
|---|---|---|---|---|
| BMI | 25.56 (23.24–27.94) | 25.13 (22.82–27.64) | 26.21 (23.67–28.37) | 0.015[*] |
| WHR | 0.95 (0.91–0.98) | 0.94 (0.91–0.98) | 0.96 (0.92–0.98) | 0.029[*] |
| WHtR | 0.54 (0.51–0.59) | 0.54 (0.50–0.58) | 0.56 (0.52–0.60) | 0.002[*] |
| LAP | 35.96 (22.79–62.78) | 30.30 (19.84–50.91) | 49.30 (32.59–75.30) | <0.001[*] |
| BRI | 4.23 (3.43–5.04) | 4.09 (3.43–5.04) | 4.49 (3.75–5.46) | 0.002[*] |
| CI | 1.28 (1.23–1.34) | 1.28 (1.22–1.32) | 1.29 (1.24–1.34) | 0.023[*] |
| VAI | 1.89 (1.17–3.15) | 1.54 (1.04–2.61) | 2.67 (1.67–3.97) | <0.001[*] |
| BAI | 26.89 (24.08–30.38) | 26.52 (23.97–29.90) | 27.59 (24.19–30.84) | 0.059 |
| AVI | 16.17 (14.38–18.43) | 15.82 (14.09–18.02) | 16.56 (14.77–19.59) | 0.001[*] |
| ABSI | 0.081 (0.078–0.085) | 0.081 (0.078–0.084) | 0.081 (0.079–0.085) | 0.146 |
| TyG index | 9.03 (8.53–9.58) | 8.84 (8.40–9.35) | 9.26 (8.86–9.88) | <0.001[*] |
| CMI | 0.65 (0.41–1.07) | 0.54 (0.35–0.89) | 0.86 (0.56–1.27) | <0.001[*] |

Notes.
[*]Denotes significance at a *P* value of < 0.05.
Abbreviations: T2D, type 2 diabetes; MCI, mild cognitive impairment; BMI, body mass index; WHR, waist-hip ratio; WHtR, waist-to-height ratio; LAP, lipid accumulation product; BRI, body roundness index; CI, conicity index; VAI, visceral adiposity index; BAI, body adiposity index; AVI, abdominal volume index; ABSI, a body shape index; TyG, triglyceride glucose; CMI, cardiometabolic index.

**Table 3** The logistic regression analyses between obesity-related indices and MCI in patients with T2D.

| Variable | Modle 1 | | Modle 2 | | Modle 3 | |
|---|---|---|---|---|---|---|
| | OR (95% CI) | *P* value | OR (95% CI) | *P* value | OR (95% CI) | *P* value |
| BMI (per 1 kg/m$^2$) | 1.055 (1.009~1.103) | 0.018 | 1.060 (1.012~1.110) | 0.014 | 1.075 (1.025~1.128) | 0.003 |
| WHR (per 0.1) | 1.289 (0.990~1.679) | 0.059 | 1.255 (0.958~1.643) | 0.099 | 1.352 (1.022~1.789) | 0.035 |
| WHtR (per 0.1) | 1.425 (1.101~1.843) | 0.007 | 1.378 (1.057~1.798) | 0.018 | 1.435 (1.090~1.890) | 0.010 |
| LAP (per 1) | 1.014 (1.009~1.019) | <0.001 | 1.015 (1.010~1.020) | <0.001 | 1.016 (1.011~1.022) | <0.001 |
| BRI (per 1) | 1.073 (1.042~1.319) | 0.008 | 1.154 (1.022~1.303) | 0.021 | 1.175 (1.035~1.333) | 0.013 |
| CI (per 0.1) | 1.282 (1.043~1.575) | 0.018 | 1.234 (0.999~1.523) | 0.051 | 1.251 (1.007~1.554) | 0.043 |
| VAI (per 1) | 1.280 (1.161~1.412) | <0.001 | 1.290 (1.167~1.425) | <0.001 | 1.294 (1.168~1.435) | <0.001 |
| BAI (per 1) | 1.031 (0.997~1.065) | 0.073 | 1.024 (0.990~1.060) | 0.166 | 1.022 (0.986~1.058) | 0.234 |
| AVI (per 1) | 1.070 (1.023~1.120) | 0.003 | 1.072 (1.023~1.123) | 0.004 | 1.091 (1.039~1.146) | 0.001 |
| ABSI (per 0.01) | 1.306 (0.936~1.822) | 0.116 | 1.219 (0.867~1.716) | 0.255 | 1.196 (0.843~1.698) | 0.316 |
| TyG index (per 1) | 2.246 (1.759~2.866) | <0.001 | 2.417 (1.869~3.125) | <0.001 | 2.608 (1.976~3.443) | <0.001 |
| CMI (per 1) | 2.248 (1.656~3.051) | <0.001 | 2.354 (1.719~3.223) | <0.001 | 2.565 (1.831~3.595) | <0.001 |

Notes.
Model 1: Unadjusted. Model 2: Adjusted for age, education, hypertension and stroke. Model 3: Model 2 plus additional adjustments for anti-diabetic drugs (GLP-1R agonists and SGLT-2 inhibitors) and HbA1c.

non-linearity = 0.637), WHR (*P* for non-linearity = 0.430), WHtR (*P* for non-linearity = 0.452), BRI (*P* for non-linearity = 0.252), AVI (*P* for non-linearity = 0.944), and TyG index (*P* for non-linearity = 0.514) and the risk of MCI in elderly patients with T2D (Figs. 3A–3F). Figures 3G, 3H, and 3I show that there was a nonlinear association (S

**Table 4  ORs (and 95% CIs) in patients with T2D according to quartiles of obesity-related indices[1].**

| Variable | Quartiles of obesity-related indices | | | | P value for trend[2] |
| | Quartile 1 | Quartile 2 | Quartile 3 | Quartile 4 | |
|---|---|---|---|---|---|
| BMI | 1.00 (reference) | 1.187 (0.713~1.976) | 1.751 (1.054~2.909) | 1.983 (1.187~3.312) | 0.004 |
| WHR | 1.00 (reference) | 0.940 (0.563~1.569) | 1.773 (1.088~2.889) | 1.519 (0.918~2.513) | 0.037 |
| WHtR | 1.00 (reference) | 1.245 (0.746~2.076) | 1.963 (1.182~3.261) | 1.903 (1.140~3.177) | 0.006 |
| LAP | 1.00 (reference) | 1.520 (0.862~2.680) | 4.369 (2.531~7.541) | 5.743 (3.282~10.050) | <0.001 |
| BRI | 1.00 (reference) | 1.287 (0.772~2.148) | 2.001 (1.202~3.331) | 1.939 (1.159~3.245) | 0.006 |
| CI | 1.00 (reference) | 1.438 (0.877~2.357) | 1.202 (0.741~1.948) | 1.627 (0.996~2.741) | 0.117 |
| VAI | 1.00 (reference) | 1.178 (0.838~2.601) | 3.714 (2.159~6.338) | 5.456 (3.125~9.443) | <0.001 |
| BAI | 1.00 (reference) | 0.744 (0.449~1.233) | 1.160 (0.702~1.917) | 1.216 (0.740~2.006) | 0.184 |
| AVI | 1.00 (reference) | 1.445 (0.869~2.404) | 1.779 (1.067~2.967) | 2.734 (1.623~4.605) | <0.001 |
| ABSI | 1.00 (reference) | 1.267 (0.778~2.063) | 1.059 (0.665~1.713) | 1.297 (0.766~2.197) | 0.454 |
| TyG index | 1.00 (reference) | 1.991 (1.136~3.489) | 3.543 (2.018~6.220) | 7.440 (4.139~13.373) | <0.001 |
| CMI | 1.00 (reference) | 2.256 (1.273~3.996) | 4.307 (2.439~7.607) | 7.355 (4.134~13.085) | <0.001 |

**Notes.**
[1] ORs and 95% CIs were calculated with the use of binary logistic regression model adjusted for age, education, hypertension, stroke, anti-diabetic drugs (GLP-1R agonists and SGLT-2 inhibitors) and HbA1c.
[2] Tests for trend were conducted by treating the quartiles as a continuous variable and assigning the median for each quartile.

shaped relation) between LAP, VAI or CMI and risk of MCI in elderly patients with T2D (all $P$ for non-linearity < 0.001). The risk of MCI increased when the level of LAP exceeded 36.11 after adjusting for age, education, hypertension, stroke, anti-diabetic drugs (GLP-1R agonists and SGLT-2 inhibitors) and HbA1c (Fig. 3G). The level of VAI (more than 1.91, Fig. 3H) and CMI (more than 0.65, Fig. 3I) also showed similar nonlinear patterns.

Figure 4 illustrates the the ROC curves and AUC values of the twelve obesity-related indices for identifying MCI among elderly patients with T2D. The AUC values for the various indices were as follows: CMI achieved the highest value of 0.682, with VAI at 0.679, TyG index at 0.673, LAP at 0.669, AVI at 0.580, WHtR and BRI both at 0.575, BMI at 0.560, CI at 0.556, WHR at 0.554, BAI at 0.547, and ABSI at 0.536. Detailed ROC curve statistics are in Table S1.

## DISCUSSION

In this study, we investigated the association between twelve obesity-related indices and MCI in adults aged 60 years and older with T2D. Our findings revealed that elevated BMI, WHR, WHtR, LAP, BRI, CI, VAI, AVI, TyG index, and CMI were significantly associated with an increased risk of MCI in elderly T2D patients. Additionally, these obesity-related indices, especially the TyG index, LAP, VAI and CMI, exhibited negative correlations with MoCA scores. Furthermore, elevated BMI, WHR, WHtR, BRI, AVI, and TyG index showed a linear positive association with the risk of MCI. In contrast, elevated LAP, VAI, and CMI demonstrated a non-linear relationship with an increased risk of MCI. Notably, CMI, VAI, TyG index, and LAP emerged as the most robust predictors of MCI in elderly T2D patients. Collectively, these results suggest that obesity-related indices, especially CMI, VAI, TyG index, and LAP, may serve as potential biomarkers for early diagnosis of MCI in

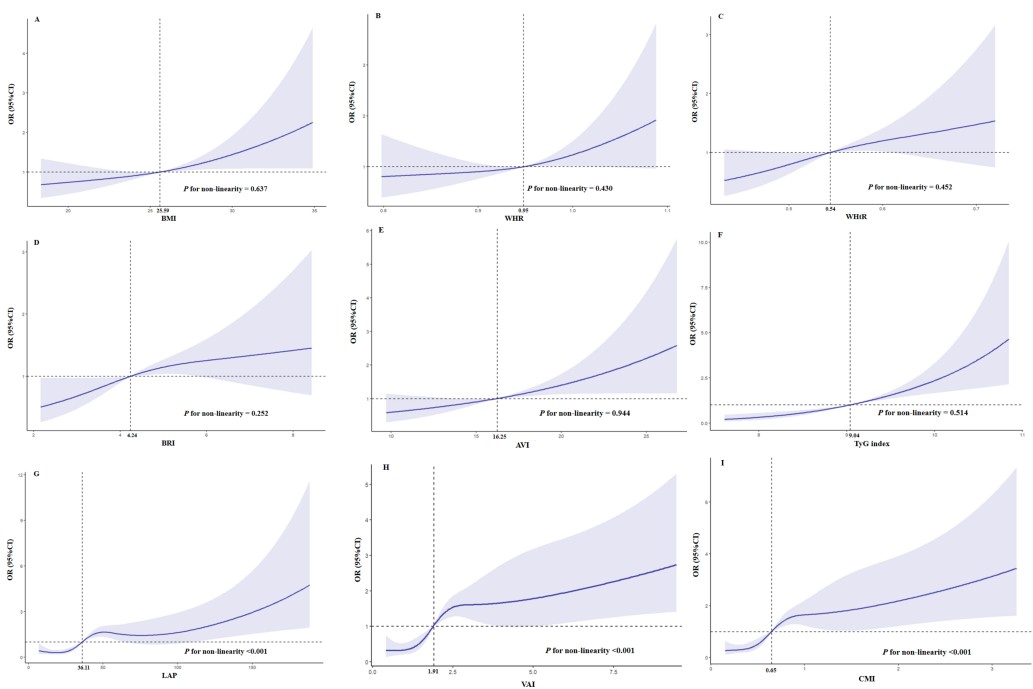

**Figure 3 Restricted cubic splines for the association obesity-related indices and MCI in elderly T2D patients.** OR of MCI by the levels of BMI (A), WHR (B), WHtR (C), BRI (D), AVI (E), TyG index (F), LAP (G), VAI (H) and CMI (I) with the use of restricted cubic splines. The analysis was adjusted for age, education, hypertension, stroke, anti-diabetic drugs(GLP-1R agonists and SGLT-2 inhibitors) and HbA1c.

elderly T2D patients. These findings provide valuable insights for future research aimed at identifying early diagnostic biomarkers of MCI in this population. Based on our thresholds, we propose that T2D patients with CMI ≥ 0.65 or VAI ≥ 1.91 or TyG index ≥ 9.04 or LAP ≥ 36.11 undergo cognitive screening using MoCA.

Obesity is an important driving force behind the cognitive impairment in T2D patients (*Feinkohl et al., 2018*). It induces cerebral microvascular dysfunction (*Balasubramanian et al., 2021*), which is a fundamental mechanism underlying T2D-associated cognitive decline (*Luo et al., 2022*; *van Sloten et al., 2020*). Interestingly, obesity and T2D share several common pathways that may contribute to cognitive impairment, such as endothelial dysfunction, blood brain barrier, neuroinflammation, gut microbiota dysbiosis, and insulin resistance (*Luo et al., 2022*; *Buie et al., 2019*; *Balasubramanian et al., 2021*). Therefore, to identify those at high risk for MCI in elderly patients with T2D, accessible and reliable indicators of obesity are urgently needed.

The association between overall obesity, as measured by BMI, and cognitive impairment remains controversial (*Grapsa et al., 2023*; *Gardener et al., 2020*; *Liang et al., 2022*). Several studies have reported a positive association between higher BMI and an increased prevalence of cognitive impairment (*Feinkohl et al., 2018*) or MCI (*Manacharoen et al., 2023*). Conversely, other studies have indicated that lower BMI may also serve as a risk factor for cognitive impairment (*Kim, Choi & Lyu, 2020*; *Dong et al., 2023*). In addition, the

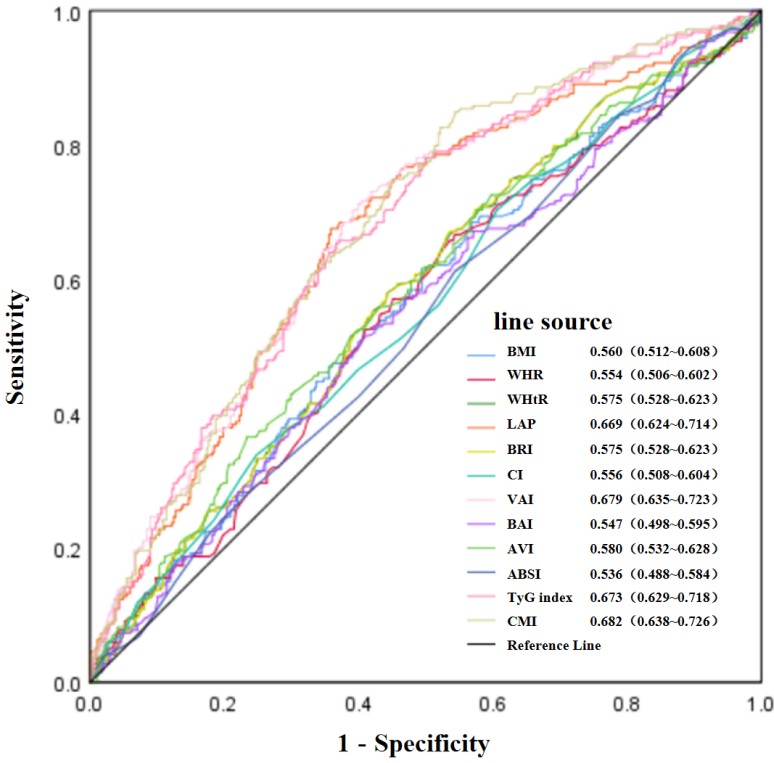

**Figure 4** Evaluation of the predictive value of obesity-related indices in identifying MCI in elderly patients with T2D.

Northern Manhattan Study did not identify any statistically significant correlation between BMI and cognitive performance (*Gardener et al., 2020*). Despite this body of research, few studies have specifically examined the relationship between BMI and cognition, particularly MCI, in patients with T2D. In our study, we observed that a higher BMI (>25.6 kg/m$^2$) was associated with an increased risk of MCI in T2D patients, with the strength of this association appearing to increase as BMI increases. However, no significant correlation was found between BMI and MoCA scores, which aligns with previous findings (*Yu et al., 2020*). Furthermore, our study suggests that certain obesity-related indices, particularly CMI, TyG index, and LAP, serve as more reliable biomarkers than BMI for screening MCI in T2D patients. One possible explanation for this finding is that BMI does not adequately capture variations in body composition and fat distribution (*Li et al., 2022*).

There is a growing body of evidence indicating that visceral adipose tissue, rather than subcutaneous adipose tissue, which has been traditionally regarded as the pathogenic adipose tissue compartment, is strongly associated with vascular risk factors and cognitive impairment (*Anand et al., 2022*; *Al-Kuraishy et al., 2023*; *Oba et al., 2022*). Obesity-related indices, such as WHR, WHtR, LAP, BRI, CI, VAI, AVI, ABSI, and CMI have been recognized as reliable markers of visceral obesity (*Li et al., 2022*; *Zhu et al., 2024*; *Hu et al., 2022*; *Nagayama et al., 2022*). Although an epidemiological study suggested that visceral obesity may play a distinct etiological role in cognitive function beyond overall

adiposity (*Mina et al., 2023*), the relationship between visceral obesity-related indices and cognitive impairment remains debated.

WHR serves as a convenient and reliable marker of visceral obesity. Some studies have demonstrated that elevated WHR is associated with cognitive impairment (*Mina et al., 2023*; *Hou et al., 2019*; *Sakib et al., 2022*). Conversely, one study reported that a higher WHR was linked to improved cognitive performance in elderly populations (*Liu et al., 2021*). LAP, another indicator of visceral adiposity, has generally been linked to cognitive impairment. A retrospective study of 220 patients with T2D revealed that LAP outperformed BMI as a screening marker for MCI and that increased LAP was an independent risk factor for MCI (*Yu et al., 2020*). However, another study found no significant association between LAP and Mini-Mental State Examination (MMSE) scores (*Huang et al., 2022*). CMI, which integrates both obesity indicators and biochemical parameters, may serve as a valuable biomarker for cognitive impairment. A cross-sectional study found that elevated CMI is associated with an increased risk of cognitive impairment in diabetic patients (*Liu et al., 2024a*). Conversely, a longitudinal study revealed that higher VAI, which provides a comprehensive assessment of visceral fat status, was linked to slower cognitive decline among Chinese populations, suggesting that increasing visceral fat might have a beneficial effect on cognition (*Zeng et al., 2023*). However, these studies seldom investigate the potential non-linear relationship between obesity-related indices and MCI. In our study, we explored the relationships between various markers of visceral obesity and MCI in elderly T2D patients and discovered that elevated LAP, VAI, and CMI were associated with an increased risk of MCI in a non-linear manner (a S-shaped relation). Specifically, an exposure-response association was observed within relatively high ranges of LAP ($\geq$ 36.11), VAI ($\geq$ 1.91), or CMI ($\geq$ 0.65). Additionally, we found that WHR, WHtR, BRI, and AVI exhibited linear associations with the risk of MCI. Our findings suggest that CMI, VAI, or LAP are superior to other obesity-related indices for screening MCI in elderly T2D patients.

The TyG index is a valuable predictor for visceral obesity in patients with T2D (*Yang et al., 2023*). Insulin resistance has been established as a central pathophysiological characteristic of T2D and is increasingly recognized as a critical contributor to the development of cognitive dysfunction (*Arnold et al., 2018*). The TyG index has emerged as a readily accessible and cost-effective surrogate marker for assessing insulin resistance in clinical populations with T2D (*Chiu et al., 2020*). A meta-analysis encompassing 12 studies demonstrated a significant association between a higher TyG index and an increased risk of cognitive decline, with the positive association being more pronounced in individuals with higher TyG index levels (*Liu et al., 2024b*). This finding aligns with our previous research (*Teng et al., 2022*). However, another study found that the TyG index was not associated with MMSE scores (*Huang et al., 2022*). Our study found that an elevated TyG index was associated with an increased risk of MCI in adults aged 60 years and older with T2D. Furthermore, our findings indicated a more robust association in the highest quartile of TyG index levels compared to the lowest quartile. Additionally, the levels of the TyG index exhibited a positive linear relationship with the risk of MCI, suggesting that elevated TyG index levels may accelerate the progression of MCI.

The primary strength of our study lies in its comprehensive examination of the association between twelve obesity-related indices and MCI, as well as its exploration of potential non-linear or linear relationships between these indices and MCI in elderly patients with T2D. However, several limitations must be acknowledged. Foremost among these is the retrospective nature of the study, which constrains our capacity to establish causal relationships. Secondly, the single-center design may introduce selection bias. Thirdly, this study did not account for other potential confounders, such as dietary habits. Finally, longitudinal data on changes in obesity-related indices over time were not available. Despite these limitations, our findings provide valuable insights into the importance of obesity-related indices in relation to MCI in elderly T2D patients. Further large-scale, prospective, multi-center studies are warranted to more accurately determine the predictive value of these indices.

## CONCLUSION

In conclusion, our analysis revealed that elevated obesity-related indices, including BMI, WHR, WHtR, BRI, AVI, and TyG index, were positively correlated with an increased risk of MCI in elderly T2D patients in a linear manner. In contrast, LAP, VAI, and CMI exhibited a non-linear relationship with MCI risk. Notably, among these indices, CMI, VAI, TyG index, and LAP emerged as the most robust predictors of MCI in adults aged 60 years and older with T2D. These findings offer a rapid, straightforward, and cost-effective method for predicting MCI in elderly T2D patients, thereby providing valuable insights for future research aimed at identifying early diagnostic biomarkers and therapeutic strategies for MCI in this population.

### Funding
This work was supported by the Hebei Province Natural Science Foundation (No. H2022307026), and the Medical Science Research Project of Hebei Province (No. 20240134). The funders had no role in study design, data collection and analysis, decision to publish, or preparation of the manuscript.

### Grant Disclosures
The following grant information was disclosed by the authors:
The Hebei Province Natural Science Foundation: No. H2022307026.
The Medical Science Research Project of Hebei Province: No. 20240134.

### Competing Interests
The authors declare there are no competing interests.

### Author Contributions
- Jing Feng conceived and designed the experiments, performed the experiments, prepared figures and/or tables, authored or reviewed drafts of the article, and approved the final draft.

- Zhenjie Teng analyzed the data, prepared figures and/or tables, and approved the final draft.
- Shuchun Chen conceived and designed the experiments, authored or reviewed drafts of the article, and approved the final draft.

## Human Ethics

The following information was supplied relating to ethical approvals (i.e., approving body and any reference numbers):

The Hebei General Hospital granted Ethical approval to carry out the study within its facilities (Ethical Application Ref: 2024-LW-109).

## Data Availability

The raw measurements are available in the Supplementary File.

## Supplemental Information

Supplemental information for this article can be found online at http://dx.doi.org/10.7717/peerj.19442#supplemental-information.

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
