# Peer review of "Associations of obesity-related indices with mild cognitive impairment in adults 60 years and older with type 2 diabetes: a retrospective study"

_PeerJ, doi:10.7717/peerj.19442_

## Round 0.1 · original submission · Minor Revisions

The authors are invited to address all the issues raised both the reviewers.

·

Basic reporting

Some more literature references can be given on the following:-
1.Anthropometric indices and surrogate markers of insulin resistance as a function of lipid ratios
2. Since thyroid dysfunction is associated with mild cognitive impairment in the elderly, better to do future studies on thyroid comorbidity in insulin resistance as a separate entity and compare the same with type 2 diabetics who are clinically euthyroid.

Experimental design

It is better to mention, in brief, the methods used in the quantitation of biochemical analytes.

Validity of the findings

Conclusion will become more robust if the authors include in their discussion a few salient points based on previous work done, if any, to cite the associations of obesity indices with mild cognitive
impairment in adults 60 years and older with type 2 diabetes, but based on the comparison between overweight and obese elderly type 2 diabetics.

Additional comments

The authors are advised to respond to the comments of the Reviewer and send the revised manuscript.

Reviewer 2 ·

Basic reporting

This study investigates the relationship between obesity indices and mild cognitive impairment (MCI) in older adults with type 2 diabetes (T2D). The findings suggest that various obesity indices, particularly the TyG index, LAP, VAI, and CMI, are significantly associated with an increased risk of MCI in this population. Therefore, these indices could be incorporated as diagnostic and prognostic biomarkers in the follow-up of patients with T2D to facilitate early intervention and mitigate progression toward dementia.
Overall, the abstract is clear and well-structured, effectively conveying the study’s objectives and key findings. However, to avoid overly generic statements, such as CMI has "superior predictive power", it would be preferable to support this assertion with data, including values for the area under the curve (AUC), odds ratios (OR), and corresponding confidence intervals (CI). Incorporating these elements would enhance the comprehension of the abstract and improve the immediate understanding of the main findings.
The introduction provides sufficient information to understand the study's context, although the explanation of the pathophysiological mechanisms linking obesity to cognitive decline could be expanded beyond a simple list.
The methods section is clearly described and detailed, with regard to the study design and procedures.
The results are well presented; however, further discussion on the impact of disease duration on MCI development would be valuable. Additionally, offering hypotheses on why certain indices exhibit a linear relationship while others do not would enhance the analysis. Moreover, the study lacks specification of threshold values at which the indices, such as CMI, achieve significant AUCs, making it difficult to determine a diagnostic or prognostic cut-off. Including these details would improve the completeness of the results section.
The conclusion is well-structured and effectively summarizes the study’s key findings. However, it could be further enriched by detailing how the obesity indices most significantly associated with MCI development could be integrated into clinical practice for risk stratification in patients with T2D. For example, specifying threshold values at which a patient should undergo cognitive assessments such as the MoCA or MMSE would enhance the practical implications of the findings.

Experimental design

The experimental design is well planned, ensuring the validity and reliability of the results, with no evident weaknesses.

Validity of the findings

The results appear valid and backed by data, with consistent conclusions. However, the discussion on clinical applicability could be further expanded.

Additional comments

no comment

---

## Round 0.2 · accepted · Accept

The authors have satisfactorily addressed all the concerns raised by the reviewers, and the paper can be published.

Please, nominate Glycated haemoglobin, instead of glycosylated haemoglobin in the paragraph "Data collection".

·

Basic reporting

Satisfactory

Experimental design

Adequate

Validity of the findings

Statistically robust

Additional comments

Since the authors have responded to the Reviewer's comments to the best possible extent, the manuscript may now be considered as worthy of publication.

Reviewer 2 ·

Basic reporting

The authors revised the manuscript and improved the incomplete information, making the study clearer and more detailed.

Experimental design

The methods section is clearly described and detailed, with regard to the study design and procedures.

Validity of the findings

Exploring the validity of indices has important clinical implications, as they can guide diagnostic workups, avoiding unnecessary tests for patients and reducing the related costs for the national health system.